# Nutritional Status and Its Detection in Patients with Inflammatory Bowel Diseases

**DOI:** 10.3390/nu15081991

**Published:** 2023-04-20

**Authors:** Beata Jabłońska, Sławomir Mrowiec

**Affiliations:** Department of Digestive Tract Surgery, Medical University of Silesia, 40-752 Katowice, Poland

**Keywords:** inflammatory bowel diseases, Crohn’ disease, ulcerative colitis, nutritional status

## Abstract

Malnutrition is an important issue in patients with inflammatory bowel diseases (IBDs) including Crohn’s disease (CD) and ulcerative colitis (UC). It is caused by altered digestion and absorption within the small bowel, inadequate food intake, and drug–nutrient interactions in patients. Malnutrition is an essential problem because it is related to an increased risk of infections and poor prognosis in patients. It is known that malnutrition is also related to an increased risk of postsurgery complications in IBD patients. Basic nutritional screening involves anthropometric parameters with body mass index (BMI) and others (fat mass, waist-to-hip ratio, muscle strength), medical history concerning weight loss, and biochemical parameters (including the Prognostic Nutritional Index). Besides standard nutritional screening tools, including the Subjective Global Assessment (SGA), Nutritional Risk Score 2002 (NRS 2002), and Malnutrition Universal Screening Tool (MUST), specific nutritional screening tools are used in IBD patients, such as the Saskatchewan Inflammatory Bowel Disease–Nutrition Risk Tool (SaskIBD-NR Tool and IBD-specific Nutritional Screening Tool). There is a higher risk of nutrient deficiencies (including iron, zinc, magnesium) and vitamin deficiencies (including folic acid, vitamin B12 and D) in IBD patients. Therefore, regular evaluation of nutritional status is important in IBD patients because many of them are undernourished. An association between plasma ghrelin and leptin and nutritional status in IBD patients has been observed. According to some authors, anti-tumor necrosis factor (anti-TNFα) therapy (infliximab) can improve nutritional status in IBD patients. On the other hand, improvement in nutritional status may increase the response rate to infliximab therapy in CD patients. Optimization of nutritional parameters is necessary to improve results of conservative and surgical treatment and to prevent postoperative complications in patients with IBDs. This review presents basic nutritional screening tools, anthropometric and laboratory parameters, dietary risk factors for IBDs, common nutrient deficiencies, associations between anti-TNFα therapy and nutritional status, selected features regarding the influence of nutritional status, and surgical outcome in IBD patients.

## 1. Introduction

Malnutrition is an important problem in patients with inflammatory bowel diseases (IBDs) including Crohn’s disease (CD) and ulcerative colitis (UC). According to the literature, the frequency of protein–energy malnutrition in patients with active IBDs is 75% [1]. Malnutrition in IBDs is caused by altered digestion and absorption within the small bowel, inadequate food intake, and drug–nutrient interactions in patients [2,3]. It is an essential problem because it is related to an increased risk of infections and poor prognosis in patients [4]. An increased risk of postsurgery complications in patients undergoing surgery has been also observed [5]. The frequency of malnutrition is higher over the course of the disease and increases with severity. Malnutrition is reported in 38.9% of CD patients in the remission phase and in 82.8% of CD patients in the active phase [6]. Casanova et al. [7], in a prospective, multicenter study involving CD and UC patients from 30 Spanish centers, indicated some predictors of malnutrition. There were self-imposed food restrictions in most of them [7]. In this study, malnutrition was reported in 16% of patients. The multivariate analysis showed the association between previous abdominal surgery, disease activity, and avoidance of some foods in the active phase with an increased risk of malnutrition [7].

The aim of our review is to present and describe basic nutritional screening tools, anthropometric and laboratory parameters, dietary risk factors for IBDs, common nutrient deficiencies, associations between anti-TNFα therapy and nutritional status, selected issues on the influence of nutritional status, and surgical outcomes in IBD patients.

## 2. Pathogenesis of IBD

IBDs are chronic nonspecific intestinal inflammatory diseases that are characterized by remission and relapse (active) phases. IBDs are related to progressive intestinal damage, leading to altered gastrointestinal function [8]. It is known that a continuing aberrant immune response to the gut microbiota plays an important role in IBD pathogenesis [9]. The IBD etiology remains largely unknown, but four main etiologic factors have been indicated, as follows: genetics, environmental factors, microbial factors, and the immune responses. All these factors interact with each other. Although the adaptive immune response plays a major role in the IBD pathogenesis, the innate immune response is also important in induction of inflammatory processes within gastrointestinal tract [8]. Environmental risk factors include smoking, poor diet, drugs, geographical factors, air pollution, and social and psychological stress. Numerous studies have shown that smoking increases the risk of CD, but decreases the risk of UC. Low levels of vitamin D are also postulated as risk factors for IBDs. A high dosage or prolonged or frequent use of nonsteroidal anti-inflammatory drugs (NSAIDs) is also related to a higher risk of both CD and UC. The role of the stress factor has also been widely described in the world literature. Stress, anxiety, and depression can worsen IBDs. It is known that air pollution related to countries’ industrialization leads to increased levels of circulating polymorphonuclear leukocytes and plasma cytokines. Changes in the human microbiome are essential risk factors for IBDs. Differences in the bacterial species between healthy and IBD bowels have been described. There is a predomination of Firmicutes and Bacteroidetes phyla producing epithelial metabolic substrates in the healthy bowel. The bacterial species present in IBD are different; Firmicutes and Bacteroidetes are lacking, and enterobacteria are abundant in CD, whereas *Clostridium* spp. are reduced and *Escherichia coli* are overrepresented in UC. Dysfunctions of innate and adaptive immune responses are crucial in IBD pathogenesis. In patients with IBDs, the nonspecific innate immune response is mediated by epithelial cells, neutrophils, dendritic cells, monocytes, macrophages, and natural killer cells. It also involves the epithelial barrier and intestinal permeability, which deteriorate in IBDs. The specific adaptative immune response is mediated by T lymphocytes, including altered Th1 immune response in CD, and the Th2 response in UC. These T cells produce large numbers of various inflammatory cytokines in IBDs [9].

In conclusion, four etiological factors are postulated to be involved in IBD pathogenesis, including genetical, environmental, microbial, and immunological elements. The altered innate and adaptative immune responses play the most important role in IBDs, but complex interaction between all the abovementioned factors is essential in IBD pathogenesis. A summary of the most common etiological factors of IBDs is presented in Figure 1.

## 3. Pathogenesis of Malnutrition in IBD Patients

IBDs are characterized by chronic progressive inflammation within the bowels that leads to intestinal structural and functional damage manifested by a wide spectrum of gastrointestinal symptoms. Several factors, including oral food restriction, maldigestion, malabsorption, chronic diarrhea leading to blood and protein loss, and intestinal bacterial overgrowth, are associated with malnutrition in IBD patients [10].

Malnutrition is observed both in UC and in CD, but protein–energy and specific nutrient malnutrition is more common in CD compared to UC due to the disease location in CD in any part of the digestive tract and, mainly, the small bowel [10].

The reduced oral food consumption in IBD patients is related to a loss of appetite due to various gastrointestinal symptoms (nausea, vomiting, abdominal pain, and diarrhea), or drug-related adverse effects (anorexia, nausea, vomiting). The intestinal absorption of numerous nutrients can also be decreased by glucocorticoids used in the therapy of IBDs. This includes phosphorus, calcium, and zinc. Sulfasalazine, as a folic acid antagonist, may lead to folate deficiency and anemia. It is known that the risk of malnutrition increases in hospitalized patients, because hospitalization and a prolonged restrictive diet during hospitalization are also related to a significant food intake reduction [10].

Malabsorption is caused by alterations in the intestinal mucosa, including altered epithelial transport and impaired epithelial integrity. This problem is most common in CD patients due to the disease location, with the ileocecal area being the most frequent. A chronic loss of water, electrolytes, blood, and proteins through the impaired intestinal integrity caused by the inflammatory process is noted in CD patients [10].

The bacterial overgrowth within the small bowel leads to impaired nutrient digestion and absorption. Production of osmotically active metabolites by intestinal microbiota additionally aggravates gastrointestinal symptoms (including abdominal discomfort and diarrhea) in IBD patients [10].

Both the abovementioned conservative management (including glucocorticoids and sulfasalazine) and surgical treatment are associated with malnutrition in IBD patients. Major surgery (including bowel resections) and postoperative complications (including short-term anastomotic leakage and long-term short bowel syndrome) are related to a reduction in oral food intake, maldigestion, and malabsorption [10]. 

All the abovementioned pathogenic mechanisms of malnutrition in IBD patients are summarized in Figure 2.

## 4. Dietary Risk Factors for IBDs 

According to the literature, there is an association between a higher risk of UC and a high intake of total fat, polyunsaturated fatty acids (PUFAs), omega-6 fatty acids, and meat, as well as an association between a higher risk of CD and a high intake of PUFAs, omega-6 fatty acids, saturated fats, and meat. High consumption of dietary fiber and fruits decreases the risk of CD, but not UC. [11,12]. It has been reported that high fruit consumption is related to a 73–80% decreased risk of CD [11]. This association was confounded by dietary fiber intake and the fact that a diet high in fruits may conversely be low in fats and meats. There is no association between total carbohydrate consumption and IBD risk, even in studies reporting an intake greater than double the recommended daily intake [11]. A systematic review by Hou et al. [12] showed a relationship between high dietary consumption of total fats, PUFAs, omega-6 fatty acids, and meat with a higher CD and UC risk. According to Hou et al. [12], high fiber and fruit intakes are related to a lower CD risk, whereas high vegetable consumption is related to lower UC risk [12]. According to prospective study by Ananthakrishnan et al. [13], long-term consumption of dietary fiber, especially from fruit sources, decreases the risk of CD, but not UC. Lautenschlager et al. [14] did not show any relationship between consumption of refined sugar and IBDs. In addition, an association between meat, artificial sweeteners, and food additives and bowel inflammation has been reported in the literature [15].

There has been limited investigation of the role of micronutrients in IBD pathogenesis. Dietary zinc has been shown to be a risk factor that could influence the risk of IBDs through effects on autophagy, the innate and adaptive immune response, and maintenance of the intestinal barrier. There was an inverse association between zinc consumption and the risk of CD, but not UC, in two large prospective cohort studies including females [11,16]. Miyaguchi et al. [17], in a prospective, randomized, controlled trial including 20 UC patents, demonstrated a positive impact of dietary zinc consumption and a Japanese diet rich in omega-3 fatty acids on UC outcomes. According to the authors, dietary zinc consumption and a Japanese diet rich in omega-3 fatty acids can induce clinical remission in patients with mild active UC [17]. A significant inverse association between dietary and supplementary vitamin D and UC and a nonsignificant reduction in the risk of CD have been reported [11]. Saadh et al. [18] used Mendelian randomization (MR) analysis to assess risk factors for IBDs. This study enrolled 458,109 participants. Univariate and multivariate MR analyses determined risk factors for IBDs. According to this study, smoking, alcohol consumption, tea intake, and vitamin D deficiency were related to a higher risk of CD (*p* < 0.05), while vegetable and fruit consumption, breastfeeding, blood zinc, and omega-3 PUFAs were related to a lower risk of CD (*p* < 0.05). Breastfeeding is generally recommended because, according to most authors, it is the optimal food for infants and it reduces the risk of IBDs [11,19]. A meta-analysis by Xu et al. [19] including 35 studies involving 7536 CD patients, 7353 UC patients, and 330,222 controls revealed a protective role of breastfeeding against the development of CD and UC [19].

In conclusion, there is an association between a higher risk of UC and a high intake of total fat, PUFAs, omega-6 fatty acids, and meat, and between a higher risk of CD and a high intake of PUFAs, omega-6 fatty acids, saturated fats, and meat. High consumption of dietary fiber and fruits decreases the risk of CD, but not UC. According to some authors, there is an inverse association between zinc consumption and the risk of CD, but not UC. Moreover, a protective role of breastfeeding against the development of CD and UC has been demonstrated. Knowledge on dietary risk factors for IBDs is essential to prevent the development of these diseases. Further investigations on the association of diet and IBD risk are required in order to prevent IBDs as well as induce clinical remission in IBD patients.

## 5. Nutritional Deficiencies in IBD Patients 

According to current ESPEN guidelines [12], the energy requirements of most IBD patients are similar to those of healthy populations (30–35 kcal/kg/day). Energy requirements can increase in patients in the active IBD phase, which is characterized by hypermetabolism and an acute inflammatory process. This is especially relevant for patients with UC [20]. Protein requirements are higher in patients with active IBDs, and consumption should be increased (to 1.2–1.5 g/kg/d in adults) relative to that recommended in the general population [20]. Higher protein requirements are not noted in IBD patients in remission. These patients have energy requirements similar to those of the general population (approximately1 g/kg/d in adult patients) [20].

According to ESPEN, there is a higher risk of micronutrient deficiencies (including zinc, magnesium, calcium, folic acid, vitamins A, B12, D, E, and K) in IBD patients. It is associated with insufficiency of the alimentary tract caused by diarrhea, malabsorption, intestinal failure, and inadequate dietary consumption secondary to anorexia accompanying disease activity. Therefore, a regular assessment of micronutrient deficiencies, at least once per year, on a clinical level as well by laboratory measurements, is recommended [20]. The British Dietetic Association also recommends a nutritional assessment of the following serum micronutrients in IBD patients: folic acid, vitamin B_12_, vitamin D, iron, zinc, magnesium, and selenium [21].

### 5.1. Common Nutrient Deficiencies in IBD Patients

#### 5.1.1. Iron

Anemia, as a result of iron deficiency, is the most common extraintestinal manifestation of IBDs, usually complicating the course of both UC and CD. Anemia is noted in 6–74% of IBD patients [11,22]. Anemia is observed more often in hospitalized IBD patients compared to IBD outpatients. In addition, it is more frequent in CD patients compared to UC patients [23,24]. ESPEN recommends iron supplementation in all IBD patients when iron deficiency anemia is present in order to correct anemia and normalize iron stores. The oral route of administration of iron is preferred as first-line management in patients with mild anemia, with clinically inactive IBDs, and without oral iron intolerance. Intravenous iron administration should be considered as the treatment of choice in patients with clinically active IBDs, with previous intolerance to oral iron, and with a lower concentration of hemoglobin (<10 g/L), and in patients requiring erythropoiesis-stimulating agents [11,20]. 

It should be known that iron is an acute-phase reactant. Therefore, interpretation of blood laboratory test results should consider inflammatory status. Serum ferritin, CRP, transferrin saturation, and serum iron concentration should be tested to determine the presence of iron deficiency, whereas hemoglobin blood concentration, blood count, and other micronutrients should be assessed to diagnose iron deficiency anemia [21]. Mijac et al. [25] reported significantly lower values of iron, total iron-binding capacity (TIBC), and ferritin accompanied by lower hemoglobin and hematocrit values, as well as lower concentrations of albumin, prealbumin, transferrin, cholesterol, and triglycerides in IBD patients compared to HSs [25]. Sturniolo et al. [26] demonstrated significantly lower plasma iron concentrations accompanied by a lower plasma selenium concentration in patients with active UC compared to patients in remission and controls. There were no significant differences between the zinc and copper values in UC patients compared to control participants in this study [26].

In conclusion, iron deficiency is common in IBD patients. Therefore, regular basic laboratory tests, including morphology of peripheral blood and serum iron concentration, need to be conducted in IBD patients. In the case of iron deficiency, iron supplementation (preferably through the oral route, if it is possible) is recommended.

#### 5.1.2. Vitamin D

According to ESPEN, patients with IBDs with active disease, in treatment with corticosteroids, or with suspected hypovitaminosis D should be monitored for serum 25(OH) vitamin D status. If needed, calcium/vitamin D supplementation is recommended to prevent low bone mineral density. These patients are vulnerable to osteopenia and osteoporosis and their complications. Osteopenia and osteoporosis should be treated according to current guidelines on osteoporosis [20]. Leslie et al. [27] reported optimal vitamin D concentrations in only 21.8% of IBD patients. The authors recommended increasing the vitamin D dosage in IBD patients already using it and introducing vitamin D supplementation in high-risk IBD patients [28]. In addition, these authors reported a positive correlation between serum 25OHD and baseline bone mineral density (BMD) for the lumbar spine, total hip, and total body (*p* < 0.05) [27]. A study by Ratajczak et al. [29], including 239 IBD patients and 45 controls, observed optimal or high vitamin D levels in only approximately 25% of all participants. The authors did not find differences in vitamin D concentration in terms of the disease extent and severity among UC patients. There were no differences concerning the disease location, behavior, or the patient’s age at the time of diagnosis in CD patients. The authors concluded that vitamin D might not be the only factor impacting BMD. In addition, these authors recommended supplementation with an increased dose of vitamin D in IBD patients compared to the healthy adult population [29]. Vitamin deficiency is a predictor for osteoporosis. It is associated with a lower level of osteocalcin, which increases patients’ risk for osteoporosis. The correct level of vitamins D and K2 is a perquisite for achieving a higher concentration of osteocalcin [30]. In a study by Lewandowski et al. [30], the abovementioned theory was partially supported by the low serum concentration of vitamin D (14.60 ng/mL) [30]. 

There are contradictory reports regarding vitamin D concentrations in patients with CD (including in the active and remission phases) [21]. Duggan et al. [31], Gilman et al. [32], Suibhne et al. [33], Dumitrescu [34], and Tan et al. [35] reported a lower vitamin D concentration in CD patients compared to HSs, whereas Grunbaum et al. [36], Bastos et al. [37], Salacinski et al. [38], Ardizzone et al. [39], and Tajika et al. [40] demonstrated similar concentrations of vitamin D in CD patients and HSs [21].

As for CD patients, vitamin D concentrations in UC patients differ in the literature data [21]. Gilman et al. [32], Dumitrescu et al. [34], and Tan et al. [35] observed lower concentrations of vitamin D in UC patients (including in the active and remission phases) compared to controls [21], whereas Grunbaum et al. [36] and Bastos et al. [37] observed similar concentrations of vitamin D in UC patients and control participants [21]. 

Concluding, it is very important to assess vitamin levels and supplement IBD patients with vitamin D, both in the active and remission phases. It is known that vitamin D has an impact on the distribution of the fecal microbiota. High vitamin D concentrations are related to higher concentrations of beneficial microbiota and lower concentrations of pathogenic microbiota [41]. Vitamin D protects the gut barrier by regulating tight junction proteins and inhibiting intestinal apoptosis [42]. Vitamin D deficiency is related to IBD activity. Supplementation of vitamin D, in order to achieve a concentration of 30 ng/mL, can be helpful to decrease disease activity [43]. Vitamin D normalization is related to a better IBD outcome, lower risk of relapse, lower risk of indications for surgery, and improved quality of life [44].

In conclusion, despite differing reports in the world literature, vitamin D deficiency is a significant problem in IBD patients. Vitamin D deficiency is related to a higher risk of osteopenia and osteoporosis. Therefore, IBD patients should be monitored for serum 25(OH) vitamin D status. In cases of vitamin D deficiency, its oral supplementation is recommended.

### 5.2. Other Nutrient Deficiencies in IBD Patients

#### 5.2.1. Magnesium

Magnesium status has been assessed in some studies [21,45,46,47,48]. Geerling et al. [47] demonstrated a significantly lower serum concentration of magnesium in CD patients in remission than control subjects, whereas another study by this author [45] demonstrated similar concentrations of magnesium in CD patients, but lower levels in UC patients (including both in the remission and active phases of the disease) in comparison with HSs. Valentini et al. [46] analyzed magnesium levels in CD patients, UC patients, and HSs, including gender subgroups. The authors showed magnesium deficiency in 30.4%, 24.2%, and 10.3% of female CD patients, UC patients, and controls, respectively, as well as in 24.2%, 17.7%, and 0.0% of male CD patients, UC patients, and controls, respectively. The highest frequency of magnesium deficiency in this study was found in CD patients. The lowest frequency of magnesium deficiency was found in controls among both females and males. This study revealed a significantly lower magnesium concentration in male UC patients compared to controls (*p* = 0.033), whereas in the female subgroups, the difference was not statistically significant [46]. A study by Gilca-Blanariu et al. [48] including 12 CD patients, 25 UC patients, and 31 HSs showed magnesium deficiency in IBD patients compared with HSs. The authors assessed hair magnesium concentration. The authors demonstrated a significantly lower hair magnesium concentration in CD patients compared to UC patients. This study also demonstrated a significantly lower hair magnesium concentration in IBD patients compared to HSs (*p* = 0.017) and significantly lower concentrations in CD compared to UC patients (*p* = 0.038). According to the authors, this difference might be related to the ileal location of the disease in CD patients, which leads to altered absorption. In addition, a statistical trend for the association between hair magnesium concentration and UC disease activity (*p* = 0.055) was reported [48]. Hussien et al. [49], in a study including 23 UC patients and 23 controls, reported a significantly lower serum magnesium in UC patients compared with HSs (*p* < 0.001) [49].

In conclusion, there are contradictory reports on the magnesium status in IBD patients in the world literature: lower or similar serum magnesium levels are reported in IBD patients. In cases of magnesium deficiency, its oral supplementation is recommended.

#### 5.2.2. Zinc

Zinc serum concentration is decreased in the IBD acute phase because of the acute-phase response due to reduced carrier protein albumin. Chronic diarrhea leading to the loss of zinc via the alimentary tract can lead to zinc deficiency in IBD patients [50]. There are differing reports on the zinc status in IBD patients in the world literature [21]. Hinks et al. [51] reported similar zinc serum concentrations in patients with active mild CD and HSs. Geerling et al. [47,52] demonstrated lower serum zinc concentrations in CD patients in remission compared with HSs. In another study by these authors [45], serum zinc concentrations were comparable in patients with active and non-active CD and controls. Valentini et al. [46] observed similar plasma zinc concentrations in CD patients in remission, UC patients, and controls. Sturniolo et al. [26] demonstrated similar plasma zinc concentrations in UC patients (in the remission and active phases) and HSs, whereas Geerling et al. [45] noted lower zinc concentrations in UC patients than HSs. A study by Dalekos et al. [53] including 75 well-nourished UC patients (32 patients with active and 43 with inactive disease) showed a significantly higher serum zinc concentration in patients both in the active and remission phase of UC compared to controls. According to the authors, this finding suggests an association of serum zinc concentration with the inflammatory process in UC [53]. Mohammadi et al. [54], in a study including 16 CD, 19 UC, and 30 control participants, reported significantly lower serum zinc levels accompanied by lower serum albumin and total protein concentrations in IBD patients compared to controls. According to the authors, lower zinc concentrations are related to more active inflammatory processes, indicating an incorrect antioxidant system in IBD patients [54]. Additionally, Hussien et al. [49] reported a significantly lower serum zinc concentration accompanied by lower magnesium and selenium levels in UC patients compared to controls (*p* = 0.018). In a study by Hengstermann et al. [55] including 167 IBD patients and 45 healthy controls, there were similar serum zinc concentrations in inactive and active IBD patients and controls. Moreover, a higher zinc concentration was reported in patients with inactive IBDs compared to controls [55]. Schneider et al. [56], in a study including 98 CD patients and 56 UC patients, observed zinc insufficiency in 11.2% of CD patients and 14.3% of UC patients, whereas copper insufficiency was detected in 20.4% of CD patients and 7.1% of UC patients [56].

Zinc serum concentration is decreased in the IBD acute phase. Chronic diarrhea leading to the loss of zinc via the alimentary tract can lead to zinc deficiency in IBD patients. There are differing reports on the zinc status in IBD patients in the world literature. Lower or similar serum zinc concentrations are reported in CD patients, whereas higher, similar, or lower serum zinc concentrations are reported in UC patients in the world literature. Based on these abovementioned literature reviews, zinc supplementation may be considered in the acute IBD phase.

#### 5.2.3. Copper

Copper is a an acute-phase reactant, and its concentration increases in the acute phase of the disease [50]. There are several reports indicating similar copper serum concentrations in IBD patients and controls [21,45,47,51] or higher copper levels in IBD patients compared to controls [21,57]. Schneider et al. [56] observed copper insufficiency in 20.4% of CD patients and 7.1% of UC patients. These authors noted that systemic inflammation inversely impacted the serum zinc and copper concentrations, leading to an increased copper/zinc ratio [56].

In conclusion, lower, similar, or higher serum copper concentrations are reported in the literature. There are no recommendations for standard monitoring and supplementation of copper.

#### 5.2.4. Selenium

Similar to zinc, selenium is decreased in the acute-phase response due to a reduction in carrier protein albumin [50]. Geerling et al. [47] reported a lower selenium concentration in CD patients in remission in comparison with HSs. Ringstad et al. [57], Wendland et al. [58], and Gentschew et al. [59] reported lower selenium concentrations in CD patients (including in the active and remission phases) and controls. Geerling et al. [45] showed similar selenium concentrations in CD patients and HSs, but lower selenium levels in UC patients compared to HSs. Significantly lower selenium levels in UC patients compared to controls were also reported by Sturniolo et al. [26] and Hussien et al. [49]. Castro Aguilar-Tablada et al. [60], in a study including 106 IBD patients (53 with UC and 53 with CD) and 30 HSs, showed significantly lower serum selenium levels in the UC patients and CD patients compared to the HSs. Moreover, a significantly lower selenium level in CD patients compared to UC patients was reported. Selenium concentrations in IBD patients were positively correlated with nutritional (total protein, albumin, prealbumin, cholinesterase, and total cholesterol) and iron-status-related (iron, hemoglobin, and hematocrit) parameters. Serum selenium and cardiovascular status were more impaired in CD compared to UC patients. In the authors’ opinion, a correct nutritional selenium status is important to establish in IBD patients to decrease the cardiovascular risk associated with an increased number of inflammation biomarkers, especially in undernourished CD patients, and is also associated with an improved nutritional and body iron status [60].

In conclusion, lower or similar selenium concentrations are reported in IBD patients. Selenium levels are positively correlated with nutritional status in IBD patients.

#### 5.2.5. Folic Acid

Folic acid indicates malabsorption in IBD patients. In addition, sulfasalazine impairs folate absorption [50]. According to most authors, there is no difference in the plasma concentration of folic acid between IBD patients and HSs, but in numerous analyzed cases, folate supplementation is already used [21,45,47,61,62]. Contrary to the abovementioned studies, Akbulut et al. [63], in a study including 103 CD patients and 103 healthy controls, reported a significantly lower serum concentration of folic acid in CD patients compared to HSs (*p* < 0.01). Yakut et al. [61], in a study including 138 IBD patients (45 CD and 93 UC patients) and 53 HSs, noted a significantly higher rate of folate deficiency (28.8%) in CD patients (28.8%) compared to controls (3.7%) (*p*  <  0.001). According to the literature data, concentrations of folic acid in UC patients and control subjects are similar [21,45,61]. Additionally, folic acid deficiency rates are similar in UC and control groups, as follows: 2.0% vs. 0% (according to Valentini et al. [46]) and 8.6% vs. 3.7% (according to Yakut et al. [61]) in UC and control groups, respectively [21]. The higher rate of folic acid deficiency in CD patients in the literature data is associated with more frequent administration of sulfasalazine, which impairs its gastrointestinal absorption [50].

In conclusion, CD patients are particularly vulnerable to folate deficiency due to the disease location and use of sulfasalazine. Lower or similar levels of folic acid are reported in CD patients. Deficiency of folate is not reported in UC patients. A number of studies assessing rates of folate deficiency in IBD patients involved patients taking the earlier-introduced folate supplementation, which could limit the impact of abovementioned findings.

#### 5.2.6. Vitamin B12

Vitamin B12 is absorbed mainly within the distal ileum. Thus, CD patients, following ileocecal resections, are particularly vulnerable to vitamin B12 deficiency [64]. There are contradictory reports on serum levels of vitamin B12, as well as its deficiency in the world literature [21]. Similar serum B12 concentrations in CD and controls were reported by Geerling et al. [47] and Yakut et al. [61]. Geerling et al. [45], Kallel et al. [62], and Akbulut et al. [63] reported significantly lower serum vitamin B12 concentrations in CD patients compared to HSs. Valentini et al. [46] showed similar vitamin B12 deficiencies in CD patients in remission (9.6%) and HSs (0.0%) [30,43]. Contrary to this report, Yakut et al. [61] demonstrated a significantly higher rate of vitamin B12 deficiency in CD patients (22.2.%) compared to controls (7.5%) [21,61]. Regarding UC, similar serum vitamin B12 concentrations, as well as rates of vitamin B12 deficiency, in UC and control groups were reported in the literature [21,45,46,61]. The greater frequency of lower serum vitamin B12 concentrations, as well as its deficiency in CD patients in the literature, is associated with the most common ileocecal location of CD (the common site of vitamin B12 absorption) in comparison to UC.

In conclusion, CD patients, following ileocecal resections, are particularly vulnerable to vitamin B12 deficiency due to absorption of vitamin B12 within the distal ileum. This is supported by literature reports. Lower or similar concentrations of vitamin B12 are reported in CD patients, whereas similar concentrations of vitamin B12 are noted in UC patients and controls. Therefore, supplementation with vitamin B12 in CD patients following ileocecal resection should be considered. The most important reports regarding the abovementioned nutrient deficiencies are summarized in Table 1.

## 6. Nutritional Screening in IBD Patients

Basic nutritional screening involves body mass index (BMI), medical history concerning weight loss, and biochemical parameters (including albumin, transferrin, prealbumin, portal vein insulin/glucagon ratio, cholesterol, glucose), measurement of nitrogen balance and protein breakdown, hematologic or immunologic parameters, including total lymphocyte count (TLC), and C-reactive protein (CRP) [6].

According to current recommendations, a BMI < 18.5 kg/m^2^, a weight loss >10–15% within six months, and serum albumin levels <30 g/L are the best markers of severe malnutrition in patients, including IBD patients [20,22,23,65,66]. According to ESPEN (the European Society of Parenteral and Enteral Nutrition) guidelines for IBDs [20], patients with IBDs should be screened for malnutrition as patients undergoing general surgery using validated nutritional screening tools [28,67]. The Nutritional Risk Score 2002 (NRS 2002) and the Malnutrition Universal Screening Tool (MUST) are particularly recommended [68,69]. ≥3 is associated with a higher risk of gastrointestinal surgery, and it is an indication for nutritional support [70]. Other universal (nonspecific for IBDs) nutritional screening tools are the following: the SGA (Subjective Global Assessment) scale and NRI (Nutritional Risk Index), Onodera’s PNI (Prognostic Nutritional Index) [6,71], and CONUT (Controlling Nutritional Status score) [72].

In addition, nutritional scores specific for patients with IBDs have been described, including the Saskatchewan Inflammatory Bowel Disease–Nutrition Risk Tool (SaskIBD-NR Tool) [73] and the IBD-specific Nutrition Self-Screening Tool (NST) [74].

In 2016, the Global Leadership Initiative on Malnutrition (GLIM) proposed a two-step assessment of the malnutrition. In the first step, various screening tools are used in order to identify “at risk” nutritional status. In the second step, the severity of malnutrition is evaluated. The following criteria for assessment of malnutrition have been postulated: three phenotypic criteria (unintentional weight loss, low BMI, and reduced muscle mass) and two etiologic criteria (reduced food consumption or assimilation, and inflammation or disease burden). According to GLIM, at least one phenotypic criterion and one etiologic criterion are required to diagnose malnutrition. GLIM proposed two categories of malnutrition severity using the phenotypic criteria for moderate (stage 1) and severe (stage 2). The etiologic criteria should be used in the guiding and assessment of nutritional intervention. Four etiologic categories have been recommended in the classification of malnutrition: chronic disease with inflammation (1), chronic disease with minimal or without inflammation (2), acute disease or injury with severe inflammation (3), and starvation including hunger/food shortage related to socio-economic or environmental factors (4) [75].

### 6.1. Specific-for-IBD Nutritional Scores

#### 6.1.1. Saskatchewan Inflammatory Bowel Disease–Nutrition Risk Tool (SaskIBD-NR Tool)

This screening tool was initially developed locally by three dietetic interns. The authors had five key criteria during development of this nutrition screening score: it had to be simple and quick, non-invasive and economical, valid and reliable, as well easy to complete, and the data used should be routinely available. It can be used in routine assessment. The SaskIBD-NR Tool assesses four components: gastrointestinal signs, loss of weight, anorexia, and restriction of food. Questions regarding the following gastrointestinal symptoms are included in the SaskIBD-NR Tool: nausea, vomiting, diarrhea, intake of nutrients, food consumption and food avoidance, and unintentional weight loss. Questions regarding symptoms reflect the activity of IBDs (active or remission). Questions on the consumption of nutrients reflect potential micronutrient deficiencies. Questions concerning weight loss reflect protein–energy malnutrition status. It is important that this score not involve BMI, as pointed out by GLIM criteria. The study by Haskey et al. [73], performed on 110 IBD outpatients, showed higher sensitivity (82.6% vs. 26.1%), specificity (97.7% vs. 87.4%), positive predictive values (90.5% vs. 35.3%), and negative predictive values (95.5% vs. 81.7%) with the SaskIBD-NR Tool compared to the MUST. In addition, the SaskIBD-NR Tool screened 21 (19.1%) patients and the MUST indicated 17 (15.5%) patients at some nutritional risk, respectively. This tool has good sensitivity but not specificity for screening nutritional risk in patients with a BMI > 25.0 kg/m^2^. Moreover, patients who are malnourished according to the GLIM criteria are not indicated using this tool. The strength of this score possibility indicates patients at nutritional risk and the need to administer early nutritional support before development of malnutrition. The traditional screening nutritional score tools indicate patients who are already malnourished. The authors concluded that the SaskIBD-NR Tool adequately detected nutritional risk in IBD patients. This tool needs further validation. According to the authors’ opinion, due to the retrospective nature of this study, the nutritional assessment might have led to recall bias. In addition, according to the authors, the SaskIBD-NR Tool should be applied in other clinical settings outside Canadian centers, including other Canadian and international gastroenterological centers, to validate it [73].

#### 6.1.2. IBD Specific Nutrition Self-Screening Tool (IBD-NST) 

This is a self-screening tool for use in adult IBD outpatients. This tool includes IBD-specific objective and subjective parameters of nutritional status with better prediction of nutritional risk compared to scores specifically used for screening malnutrition risk. IBD-NST includes BMI, loss of weight, IBD clinical signs, and nutritional concerns. It identifies patients at nutrition risk in order to introduce dietetic evaluation and nutritional support [74].

Both the abovementioned and specific-for-IBD nutritional scores are presented in Table 1.

In conclusion, nutritional assessment in IBD patients is performed using nonspecific universal nutritional screening tools as well as specific-for-IBD scores. Both of them are useful for assessment of nutritional status. IBD-specific nutritional screening tools, in addition to standard scores, involve clinical gastrointestinal symptoms and disease activity, which are important issues in IBD patients. In our opinion, an optimal nutritional screening tool should indicate patients at nutritional risk as early as possible, before the development of malnutrition, taking into consideration gastrointestinal clinical signs and food restrictions in IBD patients, to allow early nutritional support as soon as possible. Thus, both universal and specific-for-IBD nutritional screening tools should be used in clinical practice.

## 7. BMI and Other Anthropometric Parameters in IBD Patients

### 7.1. BMI

BMI is the most frequently measured anthropometric nutritional parameter. Therefore, we have reviewed and discussed the most important reports regarding BMI in IBD patients (Table 2). 

Ghoshal et al. [76] analyzed anthropometric parameters including BMI in 62 IBD patients (7 CD patients, 55 UC patients, and 42 healthy subjects (HSs). The authors reported a lower BMI in IBD patients compared to controls (19.8 [13.7–27.5] kg/m^2^ vs. 23 [17.9–27.2] kg/m^2^, *p* < 0.0001). The other anthropometric parameters were also lower in IBD patients compared to the control group, as follows: biceps: 0.3 mm [0.1–1.9] vs. 0.5 mm [0.2–1.0], *p* < 0.0001; triceps: 0.7 mm [0.2–2.9] vs. 1.2 mm [0.5–2.3], *p* < 0.0001; skinfold thickness: biceps skinfold (BSF), triceps skinfold (TSF), and mid-arm muscle circumference (MAMC): 25.9 mm [15,16,17,18,19,20,21,23,24,25,26,27,29,30,31,32,33,34,35,36,37] vs. 26.8 mm [24,25,26,27,29,30,31,32,33,34], *p* < 0.04. Moreover, lower BMI and MAMC scores accompanied by lower serum protein concentration were observed in patients with active disease compared to patients in remission despite comparable daily dietary intake between groups [76].

Similarly, another study by Mijac et al. [25] including 23 patients with active CD, 53 patients with active UC, and 30 controls showed significantly lower anthropometric nutritional parameters, such as mid-arm circumference (MAM), MAMC, BSF, TSF, subscapular skinfold (SsSF), and suprailiac skinfold measures (SiSF), in the IBD group compared to controls [25].

Additionally, Opstelten et al. [77], in a study including 165 IBD patients and 1469 controls, noted lower BMIs in IBD patients compared to the control group (24 kg/m^2^ vs. 26 kg/m^2^; *p* < 0.01) [77]. 

Unlike the above studies, a study by Principi et al. [78] including 150 IBD patients (84 CD and 66 UC patients) in clinical remission and 100 healthy control participants reported a similar BMI between groups (IBD and control groups (26.2 ± 3.6 kg/m^2^ vs. 25.6 ± 3.8 kg/m^2^; *p* = 0.21). The difference between the results of a study by Principi et al. [78] and the abovementioned studies [25,76,77] may be associated with the fact that that the study by Principi et al. [78] enrolled only IBD patients in clinical remission.

A study by Filippi et al. [79] including 54 CD patients in clinical remission >3 months and 25 controls noted similar BMIs in the analyzed groups (22.1 ± 0.5 kg/m^2^ in both groups). In contrast to BMI, the authors reported a significantly lower fat mass in CD patients compared to controls (14.4 ± 0.8 kg vs. 16.6 ± 0.9 kg; *p* < 0.05). Additionally, the other authors [21,36,45,52,76,80,81,82] observed a similar BMI in patients with CD in remission or active CD and controls [79]. 

Ueda et al. [82], in a study including 20 CD patients, 29 UC patients, and 31 controls, reported similar BMI values in CD patients, UC patients, and healthy controls (20.5 ± 0.6 kg/m^2^ vs. 20.8 ± 0.5 kg/m^2^ vs. 21.3 ± 0.4 kg/m^2^; *p* > 0.05), respectively [82].

In contrast, a study by Benjamin et al. [83] including 123 CD patients in active and remission phases and 100 controls showed significantly lower BMIs accompanied by lower fat mass (FM) in CD patients compared to healthy subjects. These differences were found both in the remission and active phases of CD [83]. 

A study by Lu et al. [84] including 150 CD patients and 254 controls reported a lower BMI accompanied by decreases in other anthropometric parameters, such as body cell mass (BCM), the BMC index, handgrip strength (HS), and the HS index, in CD patients compared to controls (*p* < 0.0001) [84].

Additionally, a study by Molnar et al. [64] including 136 CD patients and 1752 controls showed a lower BMI and fat-free mass index (FFMI) in CD patients (median BMI: 22.0 kg/m^2^ vs. 25.1 kg/m^2^; *p* < 0.0001; FFMI: 17.3 kg/m^2^ vs. 18.4 kg/m^2^; *p* = 0.0044). The authors noted low values of BMI in 21% of the patients and lower fat-free mass index (FFMI) in 30% of the patients. In addition, low BMI was not related to gender, as opposed to FFMI, which was lower more frequently in women [64].

In conclusion, there are different BMI values reported for IBD patients in the literature. A similar or lower BMI in IBD patients compared to healthy controls is reported in the world literature. So, BMI is not a specific or sensitive marker of malnutrition in IBD patients.

### 7.2. Oher Anthropometric Parameters 

Other, less common anthropometric parameters in IBDs include fat mass, waist-to-hip ratio, and muscle strength.

Regarding fat mass in active CD, Benjamin et al. [83] noted a significantly lower fat mass in active CD patients compared to HSs, whereas in the study by Cuoco et al. [85], there was no difference between CD and control groups [21]. There are contradictory reports on fat mass in CD patients in remission [21]. Lower fat mass in CD patients in remission compared to HSs was reported by Capristo et al. [86,87] and Filippi [79], whereas some other authors did not report any difference [21,45,80,84,86,88].

A multicenter study by Valentini et al. [46] including 94 CD patients, 50 UC patients, and 61 controls noted correct nutritional status in 74% of IBD patients, according to the SGA, BMI, and serum albumin concentration. In this study, fat mass was similar in CD and control participants. However, an analysis of body composition showed a lower body cell mass in CD patients (23.1 kg, 20.8–28.7; *p* = 0.021) and UC (22.6 kg, 21.0–28.0, *p* = 0.041) compared to the control group (25.0 kg, 22.0–32.5). In addition, there was a positive correlation between handgrip strength and BCM (r = 0.703, *p* = 0.001) [46].

Regarding fat mass in UC patients in remission, Valentini et al. [46] reported a higher fat mass in UC patients, whereas Capristo et al. [87] reported a similar fat mass in UC patients and controls. According to studies comparing patients with UC in active and remission phases, there was no difference in fat mass between compared groups [21,81,89].

According to the literature, there is a higher amount of visceral adipose tissue (VAT) in CD patients (including in the active and remission phases) compared to controls [21,90,91,92]. In addition, Magro et al. [92] showed a positive correlation between BMI and VAT content in CD patients. A meta-analysis by Chan et al. [93] reported an association between obesity and the risk of CD, but not UC.

A study by Buning et al. [90] including 31 women with CD in remission and 19 control women showed a significantly higher percentage of total fat mass (37 ± 10% vs. 31 ± 10%, *p* = 0.03), VAT (1885 ± 1403 mL vs. 941 ± 988 mL, *p* = 0.02), and the VAT/FM ratio (65 ± 24 mL/kg vs. 37 ± 25 mL/kg, *p* = 0.004) in CD patients compared to controls. In patients with stricturing/fistulizing CD, a higher VAT/FM ratio compared to patients without stricturing/fistulizing CD (79 ± 0.15 mL/kg vs. 63 ± 28 mL/kg, *p* = 0.067) was found. There was a relationship between a higher baseline VAT/FM ratio and increased CD activity (*p* = 0.029) [90]. There is no information regarding VAT in UC patients, and thus further investigations on this subject are needed [21,94].

According to the available literature, there is no difference in the waist-to-hip ratio between patients with CD (including in the active and remission phases) and controls [21,47,91]. Information on the waist-to-hip ratio in UC patients was not found in the literature.

There are also contradictory reports concerning muscle strength in IBD patients in the literature [21]. Geerling et al. [47] reported lower hamstring and quadricep muscle strength in men with CD in remission compared to healthy controls, but the authors did not show any difference in this parameter between female CD patients and controls. Valentini et al. [46] demonstrated lower handgrip strength in patients with CD in remission compared to controls. Wiroth et al. [95] reported lower handgrip strength in men with CD compared to healthy men, but the authors did not show any difference in this parameter between female CD and healthy patients. Lu et al. [84] and Rizzi et al. [96] reported lower handgrip strength in CD patients (including in the active and remission phases) compared to healthy controls. Geerling et al. [45] noted similar hamstring and quadriceps muscle strength in UC patients (including in the active and remission phases) compared to controls, whereas Valentini et al. [46] reported lower handgrip strength in UC patients in remission compared to control participants. 

In conclusion, there are differing reports on anthropometric parameters in IBD patients in the literature. A lower, similar, or higher fat mass in IBD patients compared to healthy controls is reported in the world literature. A lower or similar muscle strength in IBD patients and controls is reported. So, similar to BMI, these anthropometric parameters are also not specific or sensitive markers of malnutrition in IBD patients.

A summary of the most important reports regarding BMI and other anthropometric parameters is presented in Table 3.

## 8. Relationship between Selected Laboratory Results and Surgical Outcomes in IBD Patients

Albumin serum level is one of the most important and frequently measured biochemical parameters in patients with IBDs. It is a nutritional as well as inflammatory marker and it changes depending on nutritional status and clinical outcome in CD [97]. Guo et al. [98] showed that a preoperative serum albumin concentration <35 g/L was an independent risk factor for postoperative anastomotic leakage in CD patients [98]. In Zhu et al.’s [97] study, a preoperative albumin level <33.6 g/L was an independent predictor for postoperative infectious complications [97]. The authors also reported that preoperative CRP ≥ 10 mg/L was an independent predictor for postsurgery infectious complications in IBD patients. CRP is a very useful marker of inflammatory status that correlates with IBD activity. These results are comparable with those of other studies. The authors concluded that normalization of albumin and CRP levels can improve the results of surgical treatment and decrease the incidence of postoperative infectious complications [97].

Another laboratory parameter is total lymphocyte blood count, which is used along with albumin level to determine Onodera’s Prognostic Nutritional Index (PNI), which is calculated based on the serum albumin level and total lymphocyte count in the peripheral blood using the following formula: 10 × level of albumin (g/dL) + 0.005 × total lymphocyte count (/mm^3^) [99]. It is also an important nutritional and inflammatory marker in IBDs. A decreased PNI in IBDs has been reported in some studies [5,100,101]. An association between a low PNI and an increased risk of postoperative infectious complications in patients undergoing surgery for IBDs was observed [100]. According to a study by Chohno et al. [100] evaluating the association between PNI and surgical outcome in UC patients, a lower PNI may predict the prognosis of UC patients. There was a higher median PNI in patients with mortality compared to patients with no mortality (22.6 vs. 35.6) among patients undergoing total colectomy alone. The significant predictive factor for mortality in patients undergoing total colectomy alone was PNI < 24.9. Among ileal pouch–anal anastomosis (IPAA) with ileostomy, PNI < 35.5 was a predictor of pouch-related complications. PNI did not predict pouch-related complications among IPAA without ileostomy. In the authors’ opinion, PNI may be a useful indicator in decision making concerning surgical timing and procedure [100]. Maeda et al. [5] reported a higher incidence of incisional surgical site infections (SSIs) in IBD patients with a lower PNI. The frequency of incisional SSIs was 19.8%. A multivariate analysis showed that the PNI was an independent risk factor for incisional SSIs [5]. Zhou et al. [101] assessed the utility of PNI for prediction of short-term outcomes in CD patients undergoing disease-related bowel resection. In this study, patients were divided into two groups: PNI < 40 (*n* = 30) and PNI ≥ 40 (*n* = 43). Postoperative overall and infectious complications were noted significantly more frequently in patients with PNI < 40 compared to those with PNI ≥ 40 (50.0% and 46.7% vs. 23.3% and 16.3%, respectively). In the univariate analysis, BMI < 18.5 and PNI <40 were related to a higher risk of overall and infectious complications. In the multivariate analysis, only PNI < 40 was an independent predictor for infectious complications [101]. The abovementioned studies show that PNI is a useful simple predictor of postoperative complications in patients undergoing surgery for IBDs.

An association between plasma ghrelin and leptin and nutritional status in IBD patients was observed [102,103]. Ghomraoui et al. [102] noted lower ghrelin concentrations in patients with active IBDs compared to patients with inactive disease. Ghrelin levels were significantly lower in both groups (active and inactive ones) compared to the control group. In the authors’ opinion, these results indicate a possible association between hormone levels and the loss of appetite in IBD patients. In this study, nutritional status was determined through the standardized Mini-Nutritional Assessment (MNA) questionnaire and MNA score, which are correlated with plasma ghrelin and leptin in patients with active and inactive IBDs [102]. Ates et al. [103] observed higher serum ghrelin levels in patients with active IBDs compared to IBD patients in remission. In this study, high concentrations of ghrelin correlated with the disease severity and the activity markers. Moreover, ghrelin concentrations in IBD patients appositively correlated with bioelectrical impedance analysis, body composition, and anthropometric parameters. Therefore, the authors concluded that ghrelin concentration might be useful in evaluation of the disease activity and nutritional status in IBD patients [103].

In conclusion, the association between nutritional parameters, including PNI and surgical outcome, including a higher risk of postoperative complications, has been demonstrated in the world literature. Based on the findings of abovementioned studies, we can recommend standard assessing nutritional laboratory parameters (including TLC, serum albumin concentration), as well as standard preoperative calculation of PNI. In patients with decreased PNI values, we recommend preoperative nutritional support to prevent postoperative complications such as infectious complications and anastomotic leakage following bowel resections.

## 9. Influence of Anti-Tumor Necrosis Factor (Anti-TNFα) Therapy on Nutritional Status in IBD Patients 

Conventional pharmacological treatment of IBDs involves the use of sulfasalazine, mesalamine, corticosteroids, and immunosuppressants (azathioprine, methotrexate, 6-mercaptopurine, and ciclosporine). Biological therapy has been introduced as a therapy targeting one of the crucial inflammatory cytokines produced a result of altered immune response in IBDs. In this review, we focused on the association between nutritional status and biological therapy because of the leading topic of this Special Issue [104].

An altered and abnormal immune inflammatory response within the alimentary tract is typical for IBDs. Tumor necrosis factor alpha (TNFα) is the most important mediator of this disproportionate immune response. This knowledge has been used for IBD management. Biological therapies, including antibody-based drugs targeting TNFα, are very important in IBD management. They have significantly improved the treatment results in IBD patients. Infliximab is the best-studied anti-TNFα agent. Other biological agents include adalimumab, certolizumab, golimumab, and vedolizumab [104,105]. In Europe, infliximab is indicated for therapy of severe CD and UC (including CD complicated by intestinal fistulae) in adult patients not responding to conventional treatment with corticosteroids and/or immunosuppressants or with no tolerance or presence of contraindications for conventional therapy [104].

According to some authors, anti-tumor necrosis factor (anti-TNFα) therapy (infliximab) can improve nutritional status in IBD patients [106,107,108,109,110]. Nakahigashi et al. [106], in a prospective study involving 50 patients with active CD, showed that infliximab therapy was related to an improved nutritional status. Moreover, the nutritional impact of infliximab was higher in patients with malnutrition and small bowel location [106]. The authors noted limited nutritional effects of infliximab in patients with colonic involvement alone. Based on this observation, in the authors’ opinion, nutritional improvement in patients using infliximab is related to improved malabsorption within the small bowel due to alleviation of mucosal inflammation. High concentrations of TNF-α are related to intestinal inflammation in CD patients. Therefore, anti-TNF-α therapy decreases the inflammatory process and induces healing within the intestinal mucosa, leading to a reduction in malabsorption [106]. Vadan et al. [107] reported that induction and maintenance therapy with infliximab determined weight gain and corrected malnutrition in all CD patients in clinical remission [107]. On the other hand, nutritional status influences the therapeutic effect of infliximab. According to Sumi et al. [111], the response rate to infliximab therapy could be improved by optimizing nutritional status in CD patients. The authors recommend comprehensive nutritional assessment and nutritional support prior to infliximab therapy [111]. 

The mechanism of the influence of infliximab on nutritional status in IBD patients is still a matter of debate. Two main pathways are postulated, as follows: an anti-cytokine action leading to controlling the disease and a reduction in inflammation within the intestinal mucosa and an impact on TNF-α-mediated regulation of leptin levels. As mentioned, anti-inflammatory effects led to a decrease in the disease severity and a reduction in gastrointestinal symptoms (anorexia, nauseas, vomiting, diarrhea) and improve malabsorption via stimulating healing of the intestinal mucosa. This mechanism is the most important in the acute phase of IBDs. The second mechanism, related to leptin concentration, is less known. It is known that TNF-α causes cachexia (weight loss) in acute and chronic diseases. TNF-α is not only an inflammatory cytokine; it is also related to cachexia. In this pathway, a role of leptine is postulated. It could be responsible for the association between anti-TNF-α therapy and gain of appetite leading to improved nutritional status in CD patients. The role of leptin and adiponectin in the regulation of food intake has been shown: an increased concentration of leptin causes anorexia and reflects the mass of adipose tissue. In other words, body weight is regulated by adipocyte-derived leptin, and TNF-α is a mediator of inflammation-induced cachexia in CD patients. Therefore, an anti-TNF-α agent influences BMI. Reports regarding the impact of infliximab on serum leptin concentrations are contradictory; increased levels, decreased levels, and no changes have been noted. In clinical remission, a direct effect of infliximab on fat mass and body weight control mechanisms is responsible for improved nutritional status [106,107].

In conclusion, the bilateral association between anti-tumor necrosis factor therapy and nutritional status has been reported in the world literature. This therapy is related to improved nutritional status, as optimizing nutritional status can increase the response rate to infliximab. Therefore, both biological therapy and nutritional intervention are very important and may be used complementarily in IBD patients.

## 10. Conclusions

This review shows that assessment of nutritional status using standard screening tools, such as the measurement of BMI, SGA, NRS 2002, and MUST, as well as IBD-specific scores (including SaskIBD-NR Tool and IBD-specific NST), with laboratory investigations including the measurement of PNI, is very important in patients with IBDs, because many of them are undernourished. Due to malabsorption secondary to gastrointestinal insufficiency, chronic diarrhea, and the therapy used, there are different nutrient deficiencies, including iron, magnesium, zinc, and selenium, as well as vitamins (vitamin D, folic acid, vitamin B12) in IBD patients. CD patients are more vulnerable to folate and vitamin B12 deficiencies due to ileocecal location and sulfasalazine administration compared to UC patients. Therefore, regular nutritional assessment, including laboratory tests for various nutrients, is needed in IBD patients. Optimization of nutritional parameters is necessary to improve the results of conservative and surgical treatment and prevent postoperative complications in IBD patients. 

## Figures and Tables

**Figure 1 nutrients-15-01991-f001:**
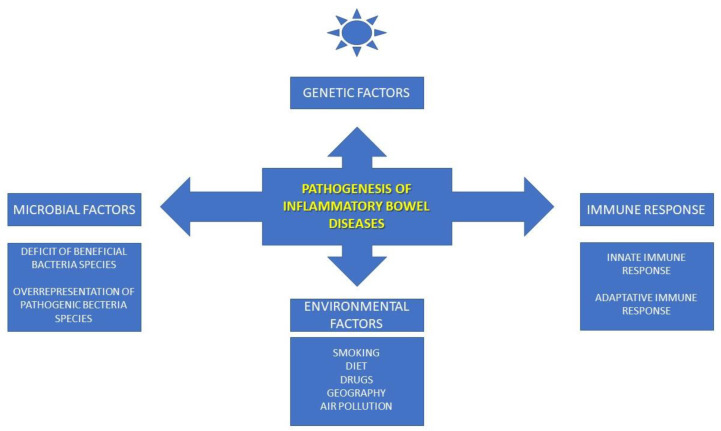
Pathogenesis of IBDs.

**Figure 2 nutrients-15-01991-f002:**
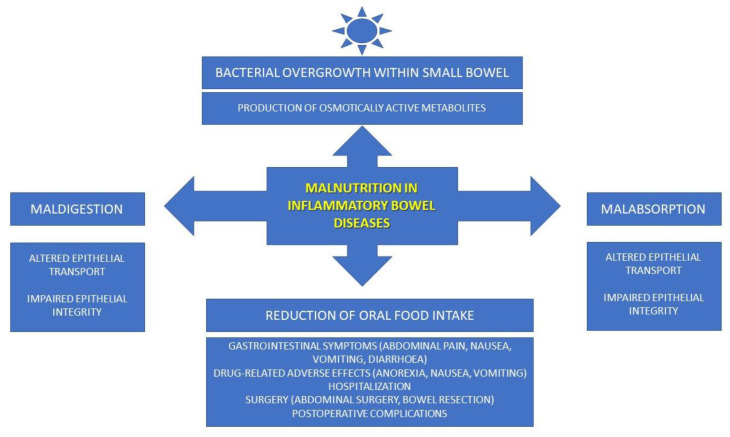
Pathomechanisms of malnutrition in IBD patients.

**Table 1 nutrients-15-01991-t001:** Summary of literature reports on nutrient deficiencies in IBD patients.

Nutrient Deficiencies in Patients with Inflammatory Bowel Diseases					
Author	Nutrient	Disease	Lower Level	Similar Level	Higher Level
Mijac et al. [25]	Iron	IBD	X		
Sturniolo et al. [26]	Iron	Active UC	X		
Duggan et al. [31]	Vitamin D	CD	X		
Gilman et al. [32]	Vitamin D	CD	X		
Suibhne et al. [33]	Vitamin D	CD	X		
Dumitrescu [34]	Vitamin D	CD	X		
Tan et al. [35]	Vitamin D	CD	X		
Grunbaum et al. [36]	Vitamin D	CD		X	
Bastos et al. [37]	Vitamin D	CD		X	
Salacinski et al. [38]	Vitamin D	CD		X	
Ardizzone et al. [39]	Vitamin D	CD		X	
Tajika et al. [40]	Vitamin D	CD		X	
Gilman et al. [32]	Vitamin D	UC	X		
Dumitrescu et al. [34]	Vitamin D	UC	X		
Tan et al. [35]	Vitamin D	UC	X		
Grunbaum et al. [36]	Vitamin D	UC		X	
Bastos et al. [37]	Vitamin D	UC		X	
Valentini et al. [46]	Magnesium	UC	X (Males)	X (Females)	
Gilca-Blanariu et al. [48]	Magnesium	IBD	X		
Hussien et al. [49]	Magnesium	UC	X		
Hinks et al. [51]	Zinc	UC		X	
Geerling et al. [47,52]	Zinc	CD	X		
Geerling et al. [45]	Zinc	CD		X	
Valentini et al. [46]	Zinc	CD, UC		X	
Sturniolo et al. [26]	Zinc	UC		X	
Geerling et al. [45]	Zinc	UC	X		
Dalekos et al. [53]	Zinc	UC			X
Mohammadi et al. [54]	Zinc	CD, UC	X		
Hussien et al. [49]	Zinc	UC	X		
Hengstermann et al. [55]	Zinc	IBD		X	
Hengstermann et al. [55]	Zinc	Inactive IBD			X
Hinks et al. [51]	Zinc	CD		X	
Geerling et al. [47]	Copper	CD		X	
Hinks et al. [51]	Copper	CD		X	
Ringstad et al. [57]	Copper	CD			X
Geerling et al. [47]	Selenium	CD	X		
Ringstad et al. [57]	Selenium	CD	X		
Wendland et al. [58]	Selenium	CD	X		
Gentschew [59]	Selenium	CD	X		
Hinks et al. [51]	Selenium	CD	X		
Hussien et al. [49]	Selenium	UC	X		
Geerling et al. [45]	Selenium	CD		X	
Sturniolo et al. [26]	Selenium	UC	X		
Hussien et al. [49].	Selenium	UC	X		
Aguilar-Tablada et al. [60]	Selenium	CD, UC	X		
Akbulut et al. [63]	Folic acid	CD	X		
Yakut al al. [61]	Folic acid	CD	X		
Geerling et al. [45]	Folic acid	UC		X	
Yakut al al. [61]	Folic acid	UC		X	
Geerling et al. [47]	Vitamin B12	CD		X	
Yakut et al. [61]	Vitamin B12	CD		X	
Geerling et al. [45]	Vitamin B12	CD	X		
Kallel et al. [62]	Vitamin B12	CD	X		
Akbulut et al. [63]	Vitamin B12	CD	X		
Geerling et al. [45]	Vitamin B12	UC		X	
Valentini et al. [46]	Vitamin B12	UC		X	
Yakut et al. [61]	Vitamin B12	UC		X	

IBD, inflammatory bowel diseases; CD, Crohn’s disease; UC, ulcerative colitis. Serum levels are compared between IBD and control healthy subjects. X, level mark.

**Table 2 nutrients-15-01991-t002:** Screening nutritional tools specific for IBDs [73,74].

Screening Tool	Questions/Points
SaskIBD-NR Tool	Have you experienced nausea, vomiting, diarrhea or poor appetite for greater than two weeks? Have you lost weight in the last month without trying? If yes, how much weight have you lost? Have you been eating poorly because of a decreased appetite? Have you been restricting any foods or food groups?
IBD-NST	Calculate your body mass index. Calculate your unplanned weight loss score. Are you currently have a flare of your symptoms? Do you have any food or nutrition concerns? Add score together. It at nutrition risk, would you like to see an IBD specialist dietitian?

IBD, inflammatory bowel disease; SaskIBD-NR Tool, Saskatchewan Inflammatory Bowel Disease–Nutrition Risk Tool; IBD-NST, IBD-specific Nutrition Self-Screening Tool.

**Table 3 nutrients-15-01991-t003:** Summary of BMI and other anthropometric parameters in IBD patients.

Author	Year	No. of Patients	Results
Ghoshal et al. [76]	2008	62 IBD (55 UC and 7 CD) patients and 42 HSs	Lower BMI, biceps, triceps, skinfold thickness, MAMC in IBD compared to control group Lower BMI, MAMC, and serum protein level in active disease compared to remission
Mijac DD et al. [25]	2010	76 IBD (23 CD and 53 UC) patients and 30 HSs	Lower anthropometric nutritional parameter scores for MAM, MAMC, TSF, BSF, SsSF, and SiSF in IBD patients compared to HSs
Opstelten JL et al. [77]	2019	165 IBD (CD and UC) patients and 1469 HSs	Lower BMI in IBD patients compared to HSs
Principi et al. [78]	2018	150 IBD (84 CD and 66 UC) patients and 100 HSs	Similar BMI in IBD patients and HSs
Filippi J et al. [79]	2006	54 CD patients (remission) and 25 HSs	Similar BMI and lower FM in CD patients compared to HSs
Ueda Y et al. [82]	2008	49 IBD (20 CD and 29 UC) patients and 25 HSs	Similar BMI in CD patients, UC patients, and HSs
Benjamin et al. [83]	2011	123 CD patients and 100 HSs	Lower BMI and FM in CD patients compared to HSs
Lu et al. [84]	2016	150 CD patients and 254 HSs	Lower BMI, BCM, BMC index, HS, HS index, in CD patients compared to HSs Lower handgrip strength in CD patients compared to HSs
Molnar et al. [64]	2017	136 CD patients and 1752 HSs	Lower BMI and FFMI in CD patients compared to HSs
Cuoco et al. [85]	2008	13 CD patients, 20 HSs	Similar FM in CD patients and HSs
Valentini al. [46]	2008	94 CD patients, 50 UC patients, 61 HSs	Similar FM in CD patients and HSs Higher FM in UC patients and HSs Lower BCM in CD and UC patients compared to HSs Positive correlation between handgrip strength and BCM Lower handgrip strength in CD and UC patients compared to HSs
Capristo et al. [87]	1998	18 CD patients, 16 UC patients, 20 HSs	Similar FM in UC patients and HSs Lower FM in CD compared to UC patients
Magro et al. [92]	2018	50 CD patients, 28 HSs	Positive correlation between BMI and VAT content in CD patients Higher VF/BMI in the CD patients compared to HSs
Chan et al. [93]	2022	563 UC patients, 1047 UC patients	Association between obesity and CD, but not UC, risk
Buning et al. [90]	2015	31 CD patients, 19 HSs	Higher FM, VAT in CD patients compared to HSs
Geerling et al. [47]	1998	32 CD patients, 32 HSs	Lower hamstring and quadricep muscle strength in CD patients compared to HSs (males) Similar hamstring and quadricep muscle strength in CD patients and HSs (females)
Wiroth et al. [95]	2005	41 CD patients, 25 HSs	Lower handgrip strength in CD patients compared to HSs (males) Similar handgrip strength in CD patients and HSs (females)
Rizzi et al. [96]	2012	78 CD patients, 75 HSs	Lower FM in CD patients compared to HSs Lower handgrip strength in CD patients compared to HSs
Geerling et al. [45]	2000	23 CD patients, 46 UC patients, 69 HSs	Lower BMI in UC patients compared to HSs Similar hamstring and quadricep muscle strength in IBD patients and HSs

IBD, inflammatory bowel disease; CD, Crohn’s disease; UC, ulcerative colitis; HSs, healthy subjects; MAMC; mid-arm muscle circumference; MAM, mid-arm circumference; TSF, triceps skinfold measures; BSF, biceps skinfold measures; SsSF, subscapular skinfold; SiSF, suprailiac skinfold measures; FM, fat mass; FFMI, fat-free mass index; BCM, body cell mass; HS, handgrip; BCM, body cell mass; VAT, visceral adipose tissue; VF, visceral fat.

## Data Availability

Not applicable.

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
