# Peer review of "Nutritional Status and Its Detection in Patients with Inflammatory Bowel Diseases"

_nutrients, 2023, doi:10.3390/nu15081991_

Round 1

Reviewer 1 Report

The review discusses the significant problem of malnutrition in patients with inflammatory bowel diseases, such as Crohn's disease and ulcerative colitis. The authors highlight the causes of malnutrition, including impaired digestive function, inadequate food intake, and drug-nutrient interactions, and the associated risks, such as a higher risk of infections and poor prognosis in patients. The authors also describe various nutritional screening tools and assessment parameters, including anthropometric and laboratory parameters, and common nutrient deficiencies.

The paper suggests that regular assessment of nutritional status is essential in IBD patients since they have an increased risk of nutrient deficits and vitamins. The authors also discuss the association between plasma ghrelin and leptin and nutritional status in patients with IBD, and how anti-tumor necrosis factor therapy can improve nutritional status in these patients. Additionally, the authors suggest that optimizing nutritional status in patients with CD can improve the response rate to infliximab treatment.

Overall, the paper provides a comprehensive review of the importance of nutritional status in patients with IBD and suggests that optimization of nutritional parameters is necessary to improve results of conservative and surgical treatment and prevent postoperative complications. The authors highlight the need for regular nutritional assessment and screening tools to identify patients at risk of malnutrition and address their nutrient deficits promptly

There are already many reviews on this subject in the literature. For example: https://www.ncbi.nlm.nih.gov/pmc/articles/PMC7071234/

https://onlinelibrary.wiley.com/doi/10.1002/jgh3.12278

https://academic.oup.com/ecco-jcc/article/5/5/443/378534

https://www.mdpi.com/2072-6643/14/4/751

ESPEN has also drawn up guidelines on this very issue.

https://www.espen.org/files/ESPEN-Guidelines/ESPEN_practical_guideline_Clinical_Nutrition_in_inflammatory_bowel_disease.pdf

This review unfortunately does not add much to what is already in the literature. It is a list of studies that can easily be found on pubmed. 

English needs to be improved. There are many paragraphs without a logical connection.

Author Response

Dear Reviewer,

Thank you for peer reviewing of our manuscript  nutrients-2346788, entitled "Nutritional Status in Patients with Inflammatory Bowel Diseases”.

Thank you for your questions and comments. We have fully addressed all the comments and my responses appear below. Our revised work includes corrections according to reviewers’ comments in the text. The changes, made according to reviewers’ comments, are marked using the „Track Changes” function in the main manuscript.

We take this opportunity to express my gratitude to the reviewers for their constructive and useful remarks. Their comments allowed us to identify areas in my manuscript that needed modification.

We also thank you for allowing me to resubmit a revised copy of the manuscript.

We hope that the revised manuscript is now acceptable for publication in Nutrients.

Responses to Reviewer 1.

Comment:

The review discusses the significant problem of malnutrition in patients with inflammatory bowel diseases, such as Crohn's disease and ulcerative colitis. The authors highlight the causes of malnutrition, including impaired digestive function, inadequate food intake, and drug-nutrient interactions, and the associated risks, such as a higher risk of infections and poor prognosis in patients. The authors also describe various nutritional screening tools and assessment parameters, including anthropometric and laboratory parameters, and common nutrient deficiencies.

The paper suggests that regular assessment of nutritional status is essential in IBD patients since they have an increased risk of nutrient deficits and vitamins. The authors also discuss the association between plasma ghrelin and leptin and nutritional status in patients with IBD, and how anti-tumor necrosis factor therapy can improve nutritional status in these patients. Additionally, the authors suggest that optimizing nutritional status in patients with CD can improve the response rate to infliximab treatment.

Overall, the paper provides a comprehensive review of the importance of nutritional status in patients with IBD and suggests that optimization of nutritional parameters is necessary to improve results of conservative and surgical treatment and prevent postoperative complications. The authors highlight the need for regular nutritional assessment and screening tools to identify patients at risk of malnutrition and address their nutrient deficits promptly

There are already many reviews on this subject in the literature. For example:

The review discusses the significant problem of malnutrition in patients with inflammatory bowel diseases, such as Crohn's disease and ulcerative colitis. The authors highlight the causes of malnutrition, including impaired digestive function, inadequate food intake, and drug-nutrient interactions, and the associated risks, such as a higher risk of infections and poor prognosis in patients. The authors also describe various nutritional screening tools and assessment parameters, including anthropometric and laboratory parameters, and common nutrient deficiencies.

The paper suggests that regular assessment of nutritional status is essential in IBD patients since they have an increased risk of nutrient deficits and vitamins. The authors also discuss the association between plasma ghrelin and leptin and nutritional status in patients with IBD, and how anti-tumor necrosis factor therapy can improve nutritional status in these patients. Additionally, the authors suggest that optimizing nutritional status in patients with CD can improve the response rate to infliximab treatment.

Overall, the paper provides a comprehensive review of the importance of nutritional status in patients with IBD and suggests that optimization of nutritional parameters is necessary to improve results of conservative and surgical treatment and prevent postoperative complications. The authors highlight the need for regular nutritional assessment and screening tools to identify patients at risk of malnutrition and address their nutrient deficits promptly

There are already many reviews on this subject in the literature. For example: https://www.ncbi.nlm.nih.gov/pmc/articles/PMC7071234/

https://onlinelibrary.wiley.com/doi/10.1002/jgh3.12278

https://academic.oup.com/ecco-jcc/article/5/5/443/378534

https://www.mdpi.com/2072-6643/14/4/751

ESPEN has also drawn up guidelines on this very issue.

https://www.espen.org/files/ESPEN-Guidelines/ESPEN_practical_guideline_Clinical_Nutrition_in_inflammatory_bowel_disease.pdf

This review unfortunately does not add much to what is already in the literature. It is a list of studies that can easily be found on pubmed. 

Answer:

Thank you for your review and comments. Thank you for detailed description of our review and opinion that our paper provides a comprehensive review of the importance of nutritional status in patients with IBD.  

Regarding presence of many reviews on this subject in the literature and some example papers, there are not similar comprehensive review in the worldwide literature. It is true that there are various original and review articles on nutritional aspects in IBD, but there are not similar review article in the worldwide literature. Thus, our paper complements the current literature on nutritional status.

Below the most important differences between pointed articles and our paper are presented:

https://www.ncbi.nlm.nih.gov/pmc/articles/PMC7071234/ - In this review, other issues (such as mechanism of malnutrition in IBD, overview on nutritional assessment, basic information on nutritional treatment in IBD, frequency of nutritional deficiencies in IBD patients) have been presented. In our paper, specific for IBD nutritional screening tools, many nutrient and vitamin deficiences, the association between nutritional status and surgical outcome, the association between anti-tumor necrosis factor therapy and nutritional status) are presented. Thus, in these two review papaer, different nutritional issues have been described.

https://onlinelibrary.wiley.com/doi/10.1002/jgh3.12278 - This is an original study, not a review of the current worldwide literature.

https://academic.oup.com/ecco-jcc/article/5/5/443/378534 - This is a recent original study, not a review of the current worldwide literature.

https://www.mdpi.com/2072-6643/14/4/751 - It is a review article is focused on dietary and nutritional approaches, not nutritional status. Our review focuses on different nutritional aspects, including nutritional screening tools and nutrient deficiences). These two papers are different in numerous points.

ESPEN has also drawn up guidelines on this very issue.

https://www.espen.org/files/ESPEN- This paper presents guidelines on clinical nutrution in IBD patients; it is not a review of the current literaturÄ™ on nutritional status in IBD. In our paper is a comprehensive review on various aspects of nutritional status in IBD. Guidelines/ESPEN_practical_guideline_Clinical_Nutrition_in_inflammatory_bowel_disease.pdf

Comments on the Quality of English Language:

English needs to be improved. There are many paragraphs without a logical connection.

Answer:

English has been improved according your suggestions. Please provide detailed, more specific comments and indicate paragraphs without a logical connection if they are still present. We will improve it.

Reviewer 2 Report

In this article, Jabłońska et al. review the importance of detecting nutritional status in IBD patients. Specifically, they review various important topics, including different screening tools, anthropometric parameters, dietary risk factors, common nutrient deficiencies in IBD, etc. The article is well-written in terms of covering major topics. Although there have been some similar review articles in the recent past, this article has unique sections avoiding the issue of lack of novelty. However, the authors need to address the following concerns for the final acceptance of the article:

  1. One of the major caveats of the article at this stage is the lack of perspectives. A review article is about more than just compiling the relevant literature. For every study or group of studies, the authors need to add perspectives, including the implication of findings, the strengths and weaknesses, potential future applications of the results, and a clear takeaway message after a section. The closest the authors have done this is in the BMI section (Lines 156-158 and lines 181-184). However, the rest of the sections merely put all the available data for the sections. They need to add perspective and conclusions to every section.
  2. The article includes a lot of text. Adding representative figures and/or tables makes the sections easier to understand.
  3. The article needs an entire section to provide a brief background about IBD pathogenesis and mainly how and why (physiology and mechanisms) the nutritional deficiencies occur.
  4. Based on the article message, a recommendation would be to rearrange sections wherein the authors first describe the different nutritional deficiencies and then talk about the screening tools.
  5. It is unclear why the authors have specifically chosen anti-TNFa therapy in IBD. Different therapies exist that have a role in improving nutritional status, including vitamin supplements and exclusive enteral nutrition (EEN). Also, the authors need to add mechanisms by which anti-TNF treatments alleviate nutritional issues.

Minor:

  1. The title needs to be inclusive: for example, “Nutritional Status and its detection in Patients with Inflammatory Bowel Diseases.”
  2. Line 120: This score needs further validation [20]. What validation does it need?
  3. Some lines have a significantly larger font size than the rest of the article.

The english language is fine with minor grammatical errors including spaces and spelling.

Author Response

Dear Reviewer,

Thank you for peer reviewing of our manuscript  nutrients-2346788, entitled "Nutritional Status in Patients with Inflammatory Bowel Diseases”.

Thank you for your questions and comments. We have fully addressed all the comments and my responses appear below. Our revised work includes corrections according to reviewers’ comments in the text. The changes, made according to reviewers’ comments, are marked using the „Track Changes” function in the main manuscript.

We take this opportunity to express my gratitude to the reviewers for their constructive and useful remarks. Their comments allowed us to identify areas in my manuscript that needed modification.

We also thank you for allowing me to resubmit a revised copy of the manuscript.

We hope that the revised manuscript is now acceptable for publication in Nutrients.

Responses to Reviewer 2.

Comment:

In this article, Jabłońska et al. review the importance of detecting nutritional status in IBD patients. Specifically, they review various important topics, including different screening tools, anthropometric parameters, dietary risk factors, common nutrient deficiencies in IBD, etc. The article is well-written in terms of covering major topics. Although there have been some similar review articles in the recent past, this article has unique sections avoiding the issue of lack of novelty. However, the authors need to address the following concerns for the final acceptance of the article:

Answer:

Thank you for your positive feedback. The manuscript has been revised and improved according to all your suggestions.

Comment:

  1. One of the major caveats of the article at this stage is the lack of perspectives. A review article is about more than just compiling the relevant literature. For every study or group of studies, the authors need to add perspectives, including the implication of findings, the strengths and weaknesses, potential future applications of the results, and a clear takeaway message after a section. The closest the authors have done this is in the BMI section (Lines 156-158 and lines 181-184). However, the rest of the sections merely put all the available data for the sections. They need to add perspective and conclusions to every section.

Answer:

Conclusions and perpectives have been added to all sections.

Comment:

  1. The article includes a lot of text. Adding representative figures and/or tables makes the sections easier to understand.

Answer:

Two figures (Figure 1. Pathogenesis of IBD – page 3, Figure 2. Pathomechanism of malnutrition in IBD patients – page 4) and one table (Table 1. Summary of literature reports on nutrient deficiencies in IBD patients – page 15) have been added. In addition, one table (Table 3. Summary of BMI and other anthropometric parameters in IBD patients = page 23) has been extended.

Comment:

  1. The article needs an entire section to provide a brief background about IBD pathogenesis and mainly how and why (physiology and mechanisms) the nutritional deficiencies occur.

Answer:

Two sections have been added: A brief background about IBD pathogenesis and pathogenic mechanisms of malnutrition in IBD patients  (pages 2-4) as follows:

2.Pathogenesis of IBD

IBD are chronic nonspecific intestinal inflammatory diseases which are characterized by remission and relapse (active) phases. IBD are related to progressive intestinal damage, leading to altered gastrointestinal function [8]. It is known that continuing aberrant immune response to the gut microbiota plays an important role in IBD pathogenesis [9]. The IBD etiology remains largely unknown, but four main etiologic factors have been indicated as follows genetic, environmental or microbial factors and the immune responses. All these factors interact with each other. Although the adaptive immune response plays a major role in the IBD pathogenesis, the innate immune response is also important in induction of inflammatory process within gastrointestinal tract [8]. The environmental risk factors involve smoking, diet, drugs, geographical factors, air pollution, as well as social and psychological stress. Numerous studies have shown that smoking increases the risk of CD, but decreases the risk of UC. Low levels of vitamin D are also postulated as risk factors for IBD. The high dosage, prolonged or frequent use of nonsteroidal anti-inflammatory drugs (NSAIDs) are also related to the higher risk both CD and UC. The role of stress factor has been also described in the worldwide literature. Stress, anxiety and depression can deteriorate IBD. It is known that air pollution, related to countries industrialization, leads to increased levels of circulating polymorphonuclear leukocytes and plasma cytokines. The changes in the human microbiome are essential risk factors in IBD. The differences in bacterial species between healthy and IBD bowels have been described. There is predomination of the Firmicutes and Bacteroidetes phyla, producing epithelial metabolic substrates, in the healthy bowel. The bacterial species in IBD are different as follows: In contrast, the Firmicutes and Bacteroidetes are lacking and enterobacteria are overgrowing in CD, whereas Clostridium spp. are reduced and Escherichia coli are overrepresented in UC. Dysfunctions of innate and adaptive immune responses are the most important in IBD pathogenesis. Patients with IBD. The non-specific innate immune response is mediated by epithelial cells, neutrophils, dendritic cells, monocytes, macrophages and natural killer cells. It also involves the epithelial barrier and intestinal permeability that are deteriorated in IBD. The specific adaptative immune response is mediated by lymphocytes T, including altered Th1 immune response in CD, and Th2 response in UC. These T cells produce large amounts various inflammatory cytokines in IBD [9].

In conclusion, four etiological factors are postulated in IBD pathogenesis, including genetical, environmental, microbial, and immunological elements. The altered innate and adaptative immune responses play the most important role in IBD, but complex interaction between all above mentioned factors is essential in IBD pathogenesis. Summary of the most common etiological factors of IBD is presented in Figure 1.

Figure 1. Pathogenesis of IBD.

3.Pathogenesis of malnutrition in IBD patients

IBD are characterized by chronic progressive inflammation within bowels that leads to the intestinal structural and functional damage manifested by a wide spectrum of gastrointestinal symptoms. Several factors, including oral food restriction, maldigestion, malabsorption, chronic diarrhea leading to blood and proteins loss, as well as intestinal bacterial overgrowth are associated with malnutrition in IBD patients [10].

Malnutrition is observed both in UC and in CD, but the protein–energy and specific nutrient malnutrition is more common in CD compared to UC due to the disease location involving any part of the digestive tract and, mainly, the small bowel in CD [10].

The reduced oral food consumption in IBD patients is related to a loss of appetite due to various gastrointestinal symptoms (nausea, vomiting, abdominal pain, and diarrhea), or drug-related adverse effects (anorexia, nausea, vomiting). The intestinal absorption of numerous nutrients can be also decreased by glucocorticoids used in the therapy of IBD. It involves: phosphorus, calcium, and zinc. Sulfasalazine, as a folic acid antagonist, may lead to folate deficiency and anemia. It is known that the risk of malnutrition increases in hospitalized patients, because hospitalization or prolonged restrictive diet during hospitalization are also related to a significant food intake reduction [10].

Malabsorption is caused by alterations of intestinal mucosa, including altered epithelial transport and impaired epithelial integrity. This problem is most common in CD patients due to the disease location with the ileocecal involvement as the most frequent. A chronic loss of water, electrolytes, blood and proteins through the impaired intestinal integrity caused by inflammatory process is noted in CD patients [10].

The bacterial overgrowth within the small bowel leads to impaired nutrients digestion and absorption. Production of osmotically active metabolites by intestinal microbiota additionally aggravates gastrointestinal symptoms (including abdominal discomfort and diarrhea) in IBD patients [10].

Both above mentioned conservative management (including glucocorticoids and sulfasalazine) and surgical treatment are associated with malnutrition in IBD patients. Major surgery (including bowel resections) and postoperative complications (including short-term anastomotic leakage and long-term short bowel syndrome) is related to reduction of oral food intake, maldigestion, and malabsorption [10].

All above mentioned pathogenic mechanisms of malnutrition in IBD patients are summarized in Figure 2.

Figure 2. Pathomechanism of malnutrition in IBD patients.

Comment:

  1. Based on the article message, a recommendation would be to rearrange sections wherein the authors first describe the different nutritional deficiencies and then talk about the screening tools.

Answer:

According to the Reviewer’s suggestion, sections have been rearranged and first the different nutritional deficiencies and then the screening tools have been presented as follows:

Section 5. Nutritional deficiencies in IBD patients (pages 10-17); Section 6. Nutritional screening in IBD patients (pages 17-23).

Comment:

  1. It is unclear why the authors have specifically chosen anti-TNFa therapy in IBD. Different therapies exist that have a role in improving nutritional status, including vitamin supplements and exclusive enteral nutrition (EEN). Also, the authors need to add mechanisms by which anti-TNF treatments alleviate nutritional issues.

Answer:

Anti-TNFa therapy in IBD has been chosen due to the subject of the Special Issue including our paper "The Role of Nutrition and Physical Activity in Autoimmune Diseases". As we have mentioned in the paper, the altered immune response plays the crucial role in the IBD pathogenesis. Therefore, we would like to review and discuss the association between nutritional status in IBD patients and biological therapy. This association is not obvious and taht is why we wanted to explore it. The influence of vitamin supplements and exclusive enteral nutrition (EEN) is logical, because this management just simply supplements nutritional deficits in IBD patients. The impact of anti-TNFa therapy on improvement of nutritional status in IBD patients is more complex.

According to the Reviewer’s suggestion, we have added mechanisms by which anti-TNF treatments alleviate nutritional issues (page 24-25) as follows:

  1. Influence of anti-tumor necrosis factor (anti-TNFα) therapy on nutritional status in IBD patients

Conventional pharmacological treatment of IBD involves as follows: sulfasalazine, mesalamine, corticosteroids and immunosuppressants (azathioprine, methotrexate, 6-mercaptopurine and ciclosporine). Biological therapy has been introduced as therapy targeting one of the crucial inflammatory cytokines produced a result of altered immune response in IBD. In this review, we focused on the association between nutritional status and biological therapy because of the leading topic of this special issue [104].

The altered and abnormal immune inflammatory response within the alimentary tract is typical for IBD. Tumor necrosis factor alpha (TNFα) a is the most important mediator of this disproportionate immune response. This knowledge was used for IBD management. Biological therapies, including antibody-based drugs targeting TNFα, are very important in the IBD management. They have significantly improved the treatment results in IBD patients. Infliximab is the best studied anti-TNFα agent, The other biological agents include adalimumab, certolizumab, golimumab, vedolizumab [104,105].In Europe, infliximab is indicated for the therapy of severe CD and UC (including CD complicated by intestinal fistulae) in adult patients without response to conventional treatment with using corticosteroids and/or immunosuppressants, or without tolerance or presence of contraindications for conventional therapy [104].

According to some authors, anti-tumor necrosis factor (anti-TNFα) therapy (infliximab) can improve nutritional status in IBD patients [106-110]. Nakahigashi et al. [106], in a prospective study involving 50 patients with active CD, showed that infliximab therapy was related to improved nutritional status. Moreover, the nutritional impact of infliximab was higher in patients with malnutrition and small bowel location, [106]. The authors noted limited nutritional effects of infliximab in patients with colonic involvement alone. Based on this observation, in authors’ opinion, nutritional improvement is patients using infliximab is related to improving malabsorption within small bowel by alleviation of mucosal inflammation. High concentrations of TNF-α are related to intestinal inflammation in CD patients. Therefore, anti-TNF-α therapy decreased the inflammatory process and induces healing within the intestinal mucosa leading to reduction of malabsorption [106].Vadan et al. [107] reported, that induction and maintenance therapy with infliximab determined weight gain and corrected malnutrition in all CD patients in clinical remission [107]. On the other hand, nutritional status influences on the therapeutic effect of infliximab. According to Sumi et al. [111], the response rate to infliximab therapy could be improved by optimizing nutritional status in CD patients. The authors recommend comprehensive nutritional assessment and nutritional support prior to infliximab therapy [111].

The mechanism of influence of infliximab on nutritional status in IBD patients is still a matter of debate. Two main pathways are postulated as follows: an anti-cytokine action leading to the disease controlling and reduction of inflammation within the intestinal mucosa and impact on TNF-α-mediated regulation of leptin levels. As the first mentioned, anti-inflammatory effect leads to a decrease of the disease severity and reduction of gastrointestinal symptoms (anorexia, nauseas, vomiting, diarrhea) and improves malabsorption via stimulating healing of the intestinal mucosa. This mechanism is the most important in the acute phase of IBD. The second mechanism, related to leptin concentration, is less known. It is known that TNF-α causes cachexia (weight loss) in acute and chronic diseases. TNF-α is not only an inflammatory cytokine, but it is also related to cachexia. In this pathway, a role of leptine is postulated. It can be responsible for the association between anti-TNF-α therapy and gain of appetite leading to improving nutritional status in CD patients. The role of leptin and adiponectin in regulation of food intake has been shown: increased concentration of leptin causes anorexia and reflects mass of adipose tissue. Using the other words, body weight is regulated by adipocyte-derived leptin, and TNF-α is a mediator of inflammation-induced cachexia in CD patients. Therefore, an anti-TNF-α agent influences on BMI. Reports regarding impact of infliximab on serum leptin concentrations are contradictory: increased levels, decreased levels and no changes were noted. In clinical remission, a direct effect of infliximab on fat mass and body weight control mechanisms are responsible for improving nutritional status [106,107].

In conclusion, the bilateral association between anti-tumor necrosis factor therapy and nutritional status has been reported in the worldwide literature. This therapy is related to improved nutritional status, and optimizing nutritional status can increase the response rate to infliximab. Therefore, both biological therapy and nutritional intervention are very important and may be used complementary in IBD patients.

Minor:

Comment:

  1. The title needs to be inclusive: for example, “Nutritional Status and its detection in Patients with Inflammatory Bowel Diseases.”

Answer:

According to the Reviewer’s, the tittle has been changed on  “Nutritional Status and its detection in Patients with Inflammatory Bowel Diseases.”

Comment:

  1. Line 120: This score needs further validation [20]. What validation does it need?

Answer:

This sentence has been presented based on the authors’ opinion. The authors presented the limitations of their study. The study was performed in the outpatient department at Royal University Hospital in Saskatoon, Saskatchewan, Canada in the Multidisciplinary Inflammatory Bowel Disease Clinic. This study was  retrospective nature and due to its retrospective nature, this assessment might  have led to recall bias. In addition, according to authors, the SaskIBD-NR Tool should be applicated also in other clinical settings, such as community based gastroenterology clinics, and in other canadian centers and internationally, to confirm the screening tool’s validity and reliability.

According to the Reviewer’s suggestion, brief summarizing explanation has been added in the manuscript (page 18) as follows:

According to authors’ opinion, due to the retrospective nature of this study, the nutritional assessment might have led to recall bias. In addition, according to authors, the SaskIBD-NR Tool should be applicated also in other clinical settings outside the Canadian centers, including other Canadian and international gastroenterological centers, to validate the SaskIBD-NR Tool [72].

Comment:

  1. Some lines have a significantly larger font size than the rest of the article.

Answer:

It has been corrected.

Comments on the Quality of English Language

The english language is fine with minor grammatical errors including spaces and spelling.

Answer:

Thank you for your positive feedback regarding English Language. Minor grammatical errors including spaces and spelling have been corrected.

Round 2

Reviewer 1 Report

The authors largely fixed the paper. 

OK!

Reviewer 2 Report

The authors have addressed all my concerns satisfactorily.